# Regulation of Tight Junctions by Sex Hormones in Goat Mammary Epithelial Cells

**DOI:** 10.3390/ani12111404

**Published:** 2022-05-30

**Authors:** Hongmei Zhu, Qianqian Jia, Yanyan Zhang, Dongming Liu, Diqi Yang, Li Han, Jianguo Chen, Yi Ding

**Affiliations:** College of Veterinary Medicine, Huazhong Agricultural University, Wuhan 430070, China; han.dong.1988@163.com (H.Z.); jia946104129@163.com (Q.J.); hzaudongyizyy@163.com (Y.Z.); liudm@mail.hzau.edu.cn (D.L.); diqiyang@mail.hzau.edu.cn (D.Y.); hanli209@mail.hzau.edu.cn (L.H.)

**Keywords:** estrogen, progesterone, tight junction, mammary gland, goat

## Abstract

**Simple Summary:**

How ovarian hormones affect goat lactation by regulating cell–cell junctions is still unclear. Through the in vivo and in vitro assays, we found that ovarian hormones could elevate cell–cell junction protein expression, which may affect the intercellular space and molecule transportation between the goat mammary epithelial cells. Our assessment suggests that ovarian hormones may affect goat milk production by regulating the cell–cell junction protein expression between mammary epithelial cells.

**Abstract:**

The sex hormones of estrogen and progesterone (P_4_) play a vital role in mammary gland development and milk lactation in ruminants. The tight junction (TJ) between adjacent secretory epithelial cells is instrumental in establishing the mammary blood–milk barrier. However, whether estrogen and P_4_ exert their effect on mammary function via regulating TJ remain unclear. Here, to clarify the role of 17-β estradiol (E_2_) and P_4_ in the regulation of TJ in goat mammary gland, we first explored the relationships between the concentrations of E_2_, P_4_, and the protein expression of claudin-1, claudin-3, occludin, and ZO-1 during the mammary gland development in goat. Then, we further explored the mRNA and protein expression of claudin-1, claudin-3, occludin, and ZO-1 in the goat mammary epithelial cells (GMECs) in vitro under different concentrations of E_2_ and P_4_. The results demonstrated that the protein expression of claudin-1 decreased, but occludin and ZO-1 increased with the decline in E_2_ and P_4_ during the transition from pregnancy to lactation. In the in vitro studies, E_2_ exerted a positive effect on the mRNA expression of claudin-1, and accelerated the proteins’ expression of claudin-1 and ZO-1 in GMECs; P_4_ upregulated the mRNA expression of claudin-1, claudin-3, occludin, and ZO-1, and also improved the protein expression of claudin-1, claudin-3, and ZO-1 in the GMECs. The results demonstrated that E_2_ and P_4_ play an important role in regulating the expression of the mammary TJ components, which may ultimately affect the mammary gland development and milk lactation.

## 1. Introduction

Mammary development begins in early fetal life, occurs slightly during puberty, while complete mammogenesis takes place during pregnancy and become fully functional after parturition. All these steps of the mammary gland development are tightly coordinated both by systemic hormones and local factors. Secretory differentiation of the alveolar mammary epithelial cells (MECs) starts around mid-pregnancy and become competent to produce and secrete some milk components at late pregnancy, while only until parturition, the milk secretion can be triggered [1]. The tight junction (TJ) is an important intercellular junctional complex that plays a vital role in controlling the epithelial barrier [2,3]. In the mammary gland, the TJ acts as both a barrier and a fence: it could prevent the paracellular transport of ions and small molecules across the cell bed (barrier function) as well as separate the cell membrane into distinct domains of protein and lipids (fence function). The mammary epithelium is leaky before lactation, while during the first days of lactation, the bidirectional paracellular exchange of molecules between the interstitial space and alveolar lumen is inhibited due to the closure of the TJ triggered by the hormonal changes. From pregnancy to lactation, the molecules can enter milk from depending on the paracellular pathways into that of the transcellular pathways to adapt to the high secretory state after parturition, which may be due to the open or closure of the tight junction (TJ) [4]. In the transition goats, the circulating electrolytes such as Na, Cl, Mg, and K were assumed to change following the different distribution of fluids in late gestation, birthing, and early lactation [5], while the loss of the TJ integrity in the goat mammary gland during lactation was closely related to milk reduction and the disturbance of the epithelia barrier [6,7,8]. Cows producing unstable milk presented elevated tight junction permeability [9,10].

Ovarian hormones of 17β-estrogen (E_2_) and progesterone (P_4_) are instrumental for mammary epithelial cell proliferation, ductal morphogenesis, mammary epithelium side branching, and alveolar formation during puberty and pregnancy [11]. There is solid evidence that glucocorticoids, prolactin, and steroids are required for the formation and maintenance of mammary TJ. Studies into whether through ovariectomy or by exogenous administration reported that P_4_ played an important role in preventing mammary TJ formation [4]. In amniotic epithelium, the P_4_/progesterone receptor (PR) pathway maintains the TJ by increasing claudin-3 and claudin-4 expression in a dose-dependent manner during mid-pregnancy [12]. Unlike P_4_, a vitro study on human primary gut tissues demonstrated that estrogen treatment decreased zonula occluden 1 (ZO-1) mRNA and protein expression, which suggested that estrogen played a role in the gut tight junction expression and permeability [13]. In the female vagina, estrogen replacement could restore the decreased expression of ZO-1, occludin, and claudin-1 to the control level after ovariectomy [14]. These studies suggest the important role of estrogen and P_4_ on modulating the expression of TJ in different tissues. However, though TJ plays an important role in maintaining the milk components in ruminant animals, studies have rarely reported on the expression pattern of TJ proteins and how ovarian hormones regulate the TJ proteins, which may affect lactation. Since the TJ structure is composed of transmembrane proteins of claudins and occludin as well as scaffolding proteins of ZO, the TJ formation or structure may vary with the change in the expression of the TJ components. Thus, in the current study, we aimed to investigate the precise function of estrogen and P_4_ on the expression of mammary TJ components during transition, which may then affect the TJ opening or closure, and finally, milk lactation. The current research will be useful in clarifying the mechanism of how E_2_ and P_4_ affect lactation through regulating the tight junction.

## 2. Materials and Methods

### 2.1. Animals and Mammary Tissue Collection

A total of fifteen female hybrids of Boer goat and Macheng black goat at the age of eight-months old (puberty) were provided by the Hubei Academy of Agricultural Sciences, Wuhan, Hubei Province, China. These goats were all healthy and on their first birth to lambs. One week was allowed for their adaption to the housing environment. Goats were mated for pregnancy. Mammary tissue and blood were collected from the goats at the age of puberty (8-months old), pregnancy day 91 (Pd91), Pd137, lactation day 4 (Ld4), and Ld31. Usually, a skin incision is made in the avascular area on the mid-abdominal line between two mammary glands and then about 1 cm^3^ mammary tissue in size was collected, which was 2–3 cm deep under the skin on one mammary gland. Tissues in different groups were collected from the same side of the mammary gland using a surgical method under general anesthesia (intravenous injection of Sumianxin, 0.01 mL/kg, Veterinary Research Institute, Jilin, China) [15]. The collected tissues were rinsed in phosphate buffered saline (PBS) for cell culture or were stored in liquid nitrogen for protein extraction. Every effort was made to reduce the number of animals used and the pain they suffered. All animal experiments were approved by the Animal Care and Use Committee of Huazhong Agricultural University (ID number: HZAURA-2015-008).

### 2.2. Preparation of Goat Mammary Epithelial Cells

The goat mammary tissues were acquired from the 8-month-old hybrids derived from Boer goats and Macheng black goats. The tissues were washed with PBS solution several times. Then, they were treated with 75% absolute ethanol and PBS, respectively. The surface layer of the tissue was removed and cut into a 1 mm^3^ block; these tissues were washed in PBS for 3–5 times until they were clean. Finally, the tissue blocks were washed with a culture medium of DMEM/F12 (Gibco, Beijing, China), and placed in 60 mm culture dishes. Four hours later, the explants were tightly attached to the dishes and 4 mL of culture medium of DMEM/F12 supplemented with 12% fetal bovine serum (FBS) was added gently into the dishes. The explants were finally cultured in a 38.5 °C, 5% CO_2_ incubator. About ten days later, both mammary epithelial cells and fibroblasts appeared in the dishes. Mammary epithelial cells could be purified from the fibroblasts after TE (0.25% trypsin/0.05% EDTA) digestion according to the different sensitivity of the cells to TE (fibroblasts are more sensitive to TE and can be digested in 1–2 min, while GMECs can be digested in 5–8 min).

### 2.3. Hormones Treatment on GMECs

Different concentrations of E_2_ (Sigma, St. Louis, MO, USA) and P_4_ (Sigma) were given to goat mammary epithelial cells (GMECs). The treatment on GMECs with a concentration of 10^−8^ M E_2_ (Sigma, St. Louis, MO, USA) and 10^−6^ M P_4_ was designated as the E_2_ + P_4_ group. The treatment on GMECs with a concentration of 1/16 × 10^−8^ M E_2_ and 10^−6^ M P_4_ was designated as the 1/16E_2_ + P_4_ group. The treatment on GMECs with concentration of 4 × 10^−8^ M E_2_ and 10^−6^ M P_4_ was designated as the 4E_2_ + P_4_ group. The treatment on GMECs with a concentration of 10^−8^ M E_2_ and 1/16 × 10^−6^ M P_4_ was designated as the E_2_ + 1/16P_4_ group. The treatment on GMECs with a concentration of 10^−8^ M E_2_ and 4 × 10^−6^ M P_4_ was designated as the E_2_ + 4P_4_ group. The treatment on GMECs with an equal volume of PBS reagent was designated as the control group. Following hormone treatment, the GMECs were harvested for extracting the mRNA and protein 24 h later.

### 2.4. ELISA

Blood was collected from the jugular vein of goats at the age of puberty (8-months old), Pd91, Pd137, Ld4, and Ld31. To acquire the serum, the blood was incubated at room temperature for 1 h, and centrifuged at 4000 rpm for 20 min. The collected serum was stored at −80 °C until use. Serum E_2_ and P_4_ were assayed with commercial ELISA kits for detecting human E_2_ and P_4_ (Beijing North Institute of Biotechnology Co. Ltd., Beijing, China), respectively. The intra- and inter-assay coefficients of variation (CV) were ≤15% for E_2_ or P_4_. The detection limits for E_2_ and P_4_ were ≤40 pg/mL and 0.2 ng/mL, respectively.

### 2.5. QRT-PCR

The RNA was extracted from the goat mammary epithelial cells by using the AG RNAex Pro Reagent (Accurate Biology, Hunan, China). Then, the total RNA was reverse transcribed into cDNA by the Thermo First cDNA Synthesis Kit (Hlingene Corporation, Shanghai, China). Quantitative real-time PCR was performed using the RealTimeFAST SYBR mix (Hlingene Corporation) to detect the gene expression of claudin-1, claudin-3, occludin, and ZO-1. The mRNA expression of β-actin served as the internal control. The primer sequences for each gene are listed in Table 1. All PCR products were cloned into the PMD18-T vector (TaKaRa, Dalian, China) for sequencing (Wuhan GeneCreate Biological Engineering Co. Ltd., Wuhan, China). The relative mRNA expression of the TJ related targets was quantified using the 2^−^^△△Ct^ method, where △△Ct = △Ct1 (experimental group)-△Ct2 (control group), and △Ct = Ct_targetgen_ − Ct_β-actin_.

### 2.6. Western Blot Analysis

The mammary tissues (collected from puberty, Pd91, Pd137, Ld4, and Ld31) or GMECs were lysed in RIPA lysis buffer (2 µL/mg, Beyotime, Shanghai, China) containing 1% protease inhibitor phenylmethanesulfonyl fluoride (PMSF) protease inhibitor. Proteins of claudin-1, claudin-3, occludin, and ZO-1 were separated using 12% SDS-PAGE gel and were transferred to polyvinylidene fluoride (PVDF) membrane (Biosharp, Wuhan, China). The dot-membranes were blocked in 5% skimmed milk for 2 h, and then were incubated with the primary antibody at 4 °C overnight. After being washed in tris buffer solution added with Tween-20 (TBST) three times, the dot-membranes were incubated with HRP-conjugated secondary antibody at room temperature for 2 h and were finally visualized through ECL (Biosharp, Wuhan, China). The primary antibodies of the rabbit anti-occludin antibody (1:1000, Affinity Biosciences, Cincinnati, OH, USA), rabbit anti-claudin-1 antibody (1:1000, ABclonal technology, Wuhan, China), rabbit anti-claudin-3 antibody (1:1000, Affinity Biosciences), rabbit anti-ZO-1 antibody (1:1000, Bioss antibodies, Beijing, China), rabbit anti-β-actin antibody (1:1000, CST, Boston, MA, USA), and secondary antibody of the HRP-conjugated Donkey anti-Rabbit IgG (1:4000, ABclonal technology, Wuhan, China) were used to detect the protein expression of occludin, claudin-1, claudin-3, and ZO-1. The quantification of the integrated optical identity (IOD) was performed by Image-Pro plus 6.0 software (Media Cybernetics, Rockville, MD, USA). The relative scales were calculated on the basis of β-actin expression.

### 2.7. Immunofluorescence

When the GMECs grew to 50–60% on the slides, immunofluorescence was performed. Briefly, the GMECs were rinsed with PBS and fixed with 4% paraformaldehyde. After being permeated with 0.3% Triton X-100 and blocked with 3% BSA, the GMECs were incubated with primary rabbit anti-CK18 antibody (1:50, Abcam, Cambridge, UK) overnight at 4 °C. Then, the coverslips were washed with PBS and incubated with secondary IFKine^TM^ Red Donkey Anti-Rabbit IgG (1:200, Abbkine, Wuhan, China) for 1 h in the dark. After being washed with PBS, the coverslips were incubated with DAPI (Beyotime, Shanghai, China) for 5 min at room temperature. Finally, the cells were washed with PBS and imaged using an EVOS fl auto imaging system (Thermo Fisher Scientific Inc, Bothell, WA, USA).

### 2.8. Statistics

Data were analyzed using SPSS 17.0 software (SPSS Inc., Chicago, IL, USA). The concentrations of E_2_ and P_4_, mRNA, and protein expression of claudin-1, claudin-3, occludin, and ZO-1 in different groups were determined with one-way ANOVA. All data were represented as means ± SD. A value of *p* < 0.05 was considered as statistically significant.

## 3. Results

### 3.1. Mammary TJ Related Targets Changed with the Alteration of E_2_ and P_4_

Female goats at the stage of puberty, Pd91, Pd137, Ld4, and Ld31 were detected with the concentrations of E_2_, and P_4_. The results from ELISA showed that the E_2_ concentration increased from Pd91 to Pd137 (*p* = 0.007), while it decreased gradually from Ld4 to Ld31 (*p* = 0.001) (Figure 1A). P_4_ concentrations decreased from Pd91 to Pd137 (*p* = 0.001) and were maintained at steady low levels on Ld4 and Ld31 (Figure 1A). The ratios of E_2_ to P_4_ were low on Pd91 and Pd137, respectively, while it increased sharply on Ld4 (*p* < 0.001) and decreased to a low value on Ld31 (*p* < 0.001) (Figure 1A). Furthermore, during the transition of pregnancy to lactation, both the E_2_ and P_4_ concentrations decreased sharply from Pd137 to Ld4 (*p* = 0.038, Figure 1A), while the ratios of E_2_ to P_4_ increased greatly on Ld4 in comparison with that on Pd137 (*p* < 0.001, Figure 1A).

To evaluate whether the levels of E_2_ and P_4_ are associated with mammary TJ protein expression, we further analyzed the protein expression of mammary TJ related targets through Western blotting (Figure 1B). Figure 1B shows that the protein expression of claudin-1 decreased from Pd137 to Ld31 (*p* = 0.036 and *p* = 0.001), while claudin-3 was the highest on Pd91 (*p* < 0.001) but then decreased to lower levels on Pd137, Ld4, or Ld31 (*p* < 0.001). Occludin decreased from Pd91 to Pd137 (*p* < 0.001), while it increased from Pd137 to Ld4 (*p* < 0.001), and then deceased on Ld31 (*p* < 0.001) (Figure 1B). ZO-1 was maintained at low levels on Pd91 and Pd137 but increased abruptly on Ld4 (*p* < 0.001), and then decreased slightly on Ld31 (*p* < 0.001). These results demonstrate that from Pd137 to Ld31, the change in claudin-1 was consistent with the change in the E_2_ and P_4_ concentrations, while occludin and ZO-1 varied with the change in the E_2_ to P_4_ ratios.

### 3.2. E_2_ and P_4_ Affected the mRNA Expression of TJ Related Targets in Goat Mammary Epithelial Cells

Mammary epithelial cells and fibroblasts were mixed together when culturing goat mammary tissues after ten days later (Figure 2A). The GMECs were purified after 3–5 passages using different digesting times, and were identified with the biochemical marker of CK-18 (Figure 2B,C).

To further explore the effects of E_2_ and P_4_ on the TJ related targets, different levels of E_2_ and P_4_ were given to GMECs to assess the mRNA expression of the TJ components. The qRT-PCR results demonstrated that the mRNA levels of claudin-1 decreased when the E_2_ concentrations increased from 1/16E_2_ to 4E_2_ (*p* < 0.001), while it increased when the P_4_ concentrations increased from 1/16P_4_ to 4P_4_ (*p* < 0.001) (Figure 3A). There was no change in the mRNA levels of claudin-3, regardless of the whether the E_2_ concentrations changed among the 1/16E_2_, E_2_, and 4E_2_, but increased when the P_4_ level was elevated from 1/16P_4_ to 4P_4_ (*p* < 0.001) (Figure 3B). Similarly, there was no change in the mRNA levels of occludin or ZO-1 when the E_2_ levels increased from 1/16E_2_ to 4E_2_, while it increased gradually when the P_4_ levels rose from 1/16P_4_ to 4P_4_ (*p* < 0.001 and *p* < 0.001, respectively) (Figure 3C,D). These results indicate that E_2_ and P_4_ play different roles in regulating the TJ mRNAs and that P_4_ dominates the upregulation of the TJ components in the mRNAs.

### 3.3. E_2_ and P_4_ Altered Protein Expression of TJ Related Targets in Goat Mammary Epithelial Cells

Then, to assess whether E_2_ and P_4_ also affected the TJ protein expression, the expression of claudin-1, claudin-3, occludin, and ZO-1 in the GMECs treated with different levels of E_2_ and P_4_ were identified with Western blotting analysis. As shown in Figure 4, the protein expression of claudin-1 was elevated when the E_2_ concentrations increased from 1/16E_2_ to 4E_2_ (*p* = 0.022). Similarly, when the P_4_ levels increased from 1/16P_4_ to 4P_4_, the protein expression of claudin-1 was greatly enhanced (*p* < 0.001). However, there was no change in the protein expression of claudin-3 when the E_2_ concentrations were 1/16E_2_, E_2_, and 4E_2_, but it was elevated significantly when the P_4_ levels increased from 1/16P_4_ to 4P_4_ (*p* < 0.001). Unexpectedly, occludin did not change whenever E_2_ or P_4_ was altered. Surprisingly, the protein expression of ZO-1 was improved whenever the E_2_ levels increased from 1/16E_2_ to 4E_2_ (*p* < 0.001) or the P_4_ levels increased from 1/16P_4_ to 4P_4_ (*p* < 0.001). These results suggest that E_2_ and P_4_ accelerated the protein expression of the TJ related targets, and that P_4_ played a dominant role in upregulating the TJ proteins.

## 4. Discussion

As a mammary blood–milk barrier and a selective gate, the tight junction separates the milk-containing compartments from the surrounding extracellular fluid and vasculature, while impairment of the mammary TJ integrity is reported to be associated with decreased milk secretion during an extended milking interval [6]. Reproductive hormones of E_2_ and P_4_ were reported in the regulation of TJ in the tissues of human gut, rat vagina, or mice amniotic by affecting the expression of ZO-1, occludin, or claudins [12,13,14]. In the current study, to investigate the role of estrogen and P_4_ in goat mammary permeability, we analyzed the expression of the TJ related targets of claudin-1, claudin-3, occludin, and ZO-1 in the goat mammary tissue as well as in the goat mammary epithelial cells. We found that the protein expression of claudin-1 decreased, while the occludin and ZO-1 increased in the goat mammary gland during the decline in the E_2_ and P_4_ concentrations from the transition of pregnancy to lactation in vivo. The mRNA and protein expression of claudin-1, claudin-3, and ZO-1 changed with the P4 alteration in GMECs in vitro. While the mRNA expression of claudin-3, occludin, and ZO-1 did not change with the alteration in E_2_, but the protein expression of claudin-1 and ZO-1 increased slightly with the E_2_ improvement in the GMECs in vitro (Figure 5). This may demonstrate that it was P_4_, which played a key role in regulating the TJ related targets of claudin-1, claudin-3, and ZO-1 in vitro, but other factors may involve in the regulation of TJ proteins in vivo. During mid- to late-pregnancy, due to the elongation of mammary ducts and the formation of the alveolar, there are great changes in the cellular sub-populations such as epithelial cell proliferation and differentiation under the control of estrogen and P_4_. These changes may lead to the reassembly of the TJ. While in mice, claudin-1 is clearly localized to the tight junctions in mammary ducts in non-pregnant and non-lactating animals; claudin-3 is localized both to the tight junction and basolateral membrane, and claudin-3 mRNA was present at the highest levels in mid-pregnancy, while it fell by about 10-fold at the onset of lactation [16]. Corresponding with these results, the current study on goats showed the decline in claudin-1 during pregnancy and lactation, while claudin-3 reached high levels in mid-pregnancy and fell to low levels in late pregnancy. Occludin has an important capacity in the assembly of the TJ and is associated with a number of diseases due to the disruption of occludin and the increase in paracellular permeability [17]. ZO-1 is a junctional component known to directly interact with occludin and is required for the TJ strand assembly [18,19]. Thus, during late pregnancy when the TJs assemble, the formation for the alveolar may lead to the decreased expression of occludin and ZO-1, while in early lactation (Ld4), the TJ of the mammary gland would have assembled and closed to increase milk production. Therefore, the protein expression of occludin and ZO-1 expression increased abruptly on Ld4.

Previous studies have reported that E_2_ ameliorates tight junction disruption via repressing the MMP transcription in microvessel endothelial bEnd.3 cells that suffered from oxygen and glucose deprivation/reperfusion injury [20], and E_2_ treatment restored the azoxymethane or dextran sodium sulfate induced mRNA inhibition of ZO-1, occludin, and claudin-4 in mice [21]. These results were consistent with the current study that E_2_ increased the protein expression of ZO-1 and claudin-1 in GMECs in vitro. However, in young ovariectomized female rats, E_2_ treatment increased the expression of occludin over the placebo, while in middle-aged rats, E_2_ decreased the protein expression of occludin in the brain [22]. However, in human primary gut tissues, estrogen decreases the ZO-1 expression [13]. In the current study, the protein levels of ZO-1 and claudin-1 increased, but occludin and claudin-3 did not change in the GMECs after imposing E_2_ treatment. These differences in E_2_ regulation may be due to the different tissues and concentrations of E_2_, or the time of hormone reaction. However, the mRNA levels of claudin-1 and ZO-1 did not change with the E_2_ alteration in the GMECs, but the protein levels of these two targets increased in vitro, which may suggest that post-transcriptional modification occurred in the E_2_ effect on claudin-1 and ZO-1 expression.

P_4_ contributes to the antenatal formation and the disappearance of TJs in uterine and mammary epithelial tissues. It is reported that progesterone receptor antagonist RU-486 administration increased the permeability of the amniotic membrane and adversely affected the localization and expression of claudin-3 and claudin-4 in the amniotic epithelium, while P_4_ administration induced increases in the claudin-3 and claudin-4 expression in a dose-dependent manner, but did not influence their localization in organ-cultured amniotic membranes [12]. These results coincided with our study that the mRNA and protein expression of claudin-3 was elevated with the increase in P_4_ concentration. Furthermore, P_4_ could block the thrombin-induced decrease in the TJ proteins of occludin, claudin-5, and ZO-1 in the mouse brain endothelial cells [23]. Physiologic concentrations of the P_4_ treatment increased the trans-epithelial electrical resistance (TEER) in the primary colon tissues and Caco-2 cells in vitro through upregulating the occludin expression [24]. In the current study, the P_4_ increase also elevated the mRNA and protein expression of claudin-1, claudin-3, and ZO-1 in the GMECs, which may indicate that P_4_ may also play a protective role in goat mammary permeability.

In the current study, the experiments of administrating E_2_ and P_4_ on the epithelial cells in vitro had many advantages such as relatively simple influencing factors and easy to control. However, since the internal environment of the body is regulated by a complex network of the nervous system, endocrine system, and immune system, it is difficult in the in vitro cell experiments to simulate the environment in vivo and may not fully reflect the situation of the hormones on the mammary gland TJ expression. Thus, a further experiment in vivo is required to verify the function of sex hormones on TJs. In dairy cows in late lactation, E_2_ injections decreased the milk yield but increased the lactose in plasma and urine, showing the effect of E_2_ on the integrity of the mammary tight junctions [25]. Nguyen verified the hypothesis by injecting [^14^C] sucrose into the lumen of mouse mammary gland where P_4_ withdrawal at the termination of pregnancy triggered mammary TJ closure [4]. These studies demonstrate that E_2_ and P_4_ play important roles in regulating the mammary TJ and milk secretion in vivo. The current study is the first to show the accurate effect of E_2_ and P_4_ on the mammary TJ expression through in vivo and in vitro studies, which would offer valuable information to explore the mechanisms of the reproductive hormones regulating goat mammary gland development and lactation.

## Figures and Tables

**Figure 1 animals-12-01404-f001:**
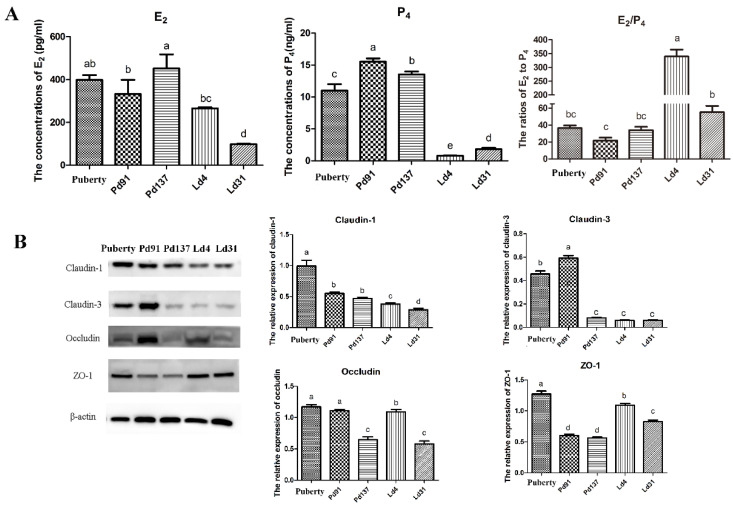
The serum E_2_ and P_4_ levels and the protein expression of the TJ proteins of goats at the stages of 8 m, Pd91, Pd137, Ld4, and Ld31. (**A**) Serum E_2_ and P_4_ levels and the ratios of E_2_ to P_4_. (**B**) The protein expression of claudin-1, claudin-3, occludin, and ZO-1 in goat mammary glands. Data are presented as the means ± SD, different letters indicate the hormones are different (*p* < 0.05).

**Figure 2 animals-12-01404-f002:**
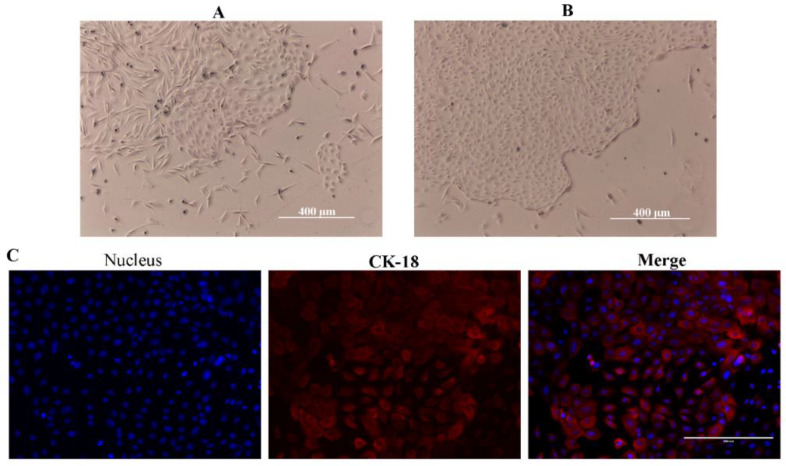
The isolation of goat mammary epithelial cells. (**A**) MECs and fibroblasts were emigrated from the cultured mammary tissue. (**B**) Purified MECs. (**C**) The immunofluorescence identification of MECs through the biochemical marker of CK-18.

**Figure 3 animals-12-01404-f003:**
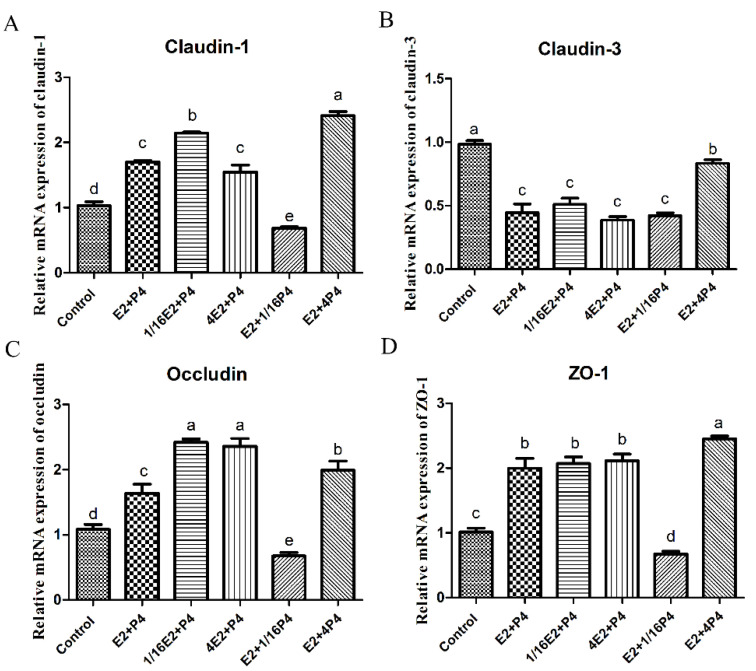
The effect of the E_2_ and P_4_ treatment on the mRNA expression of claudin-1, claudin-3, occludin, and ZO-1. (**A**) Claudin-1 mRNA expression. (**B**) Claudin-3 mRNA expression. (**C**) Occludin mRNA expression. (**D**) ZO-1 mRNA expression. These experiments were repeated at least three times. Data are represented as means ± SD; the values with different letters differed significantly (*p* < 0.05).

**Figure 4 animals-12-01404-f004:**
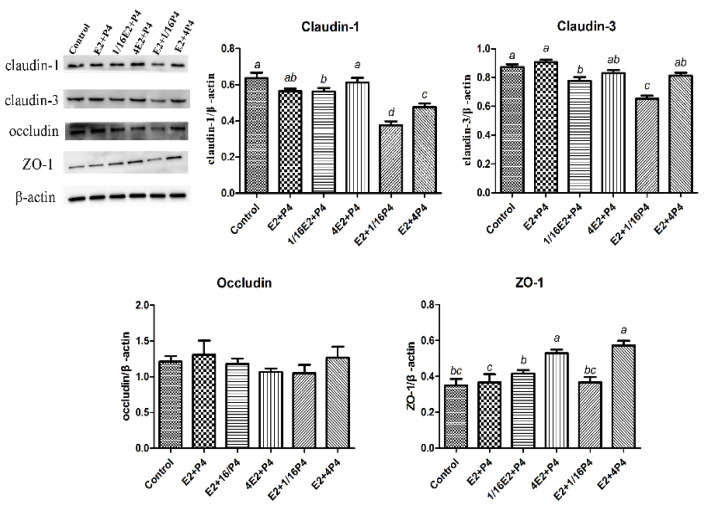
The effect of the E_2_ and P_4_ treatment on the protein expression of claudin-1, claudin-3, occludin, and ZO-1. Western blot analysis and densitometry quantification of claudin-1, claudin-3, occludin, and ZO-1 expression normalized to the total β-actin in the GMECs. These experiments were repeated at least three times. Data are represented as the means ± SD; values with different letters differed significantly (*p* < 0.05).

**Figure 5 animals-12-01404-f005:**
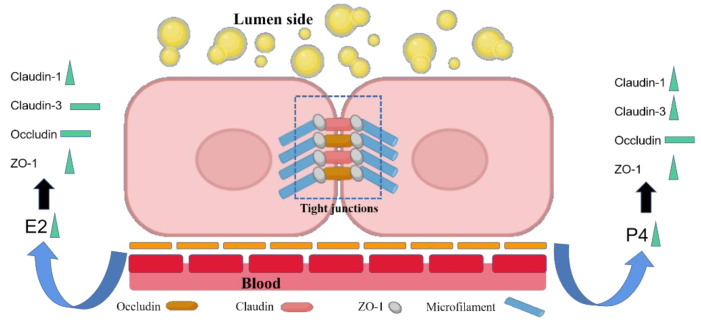
The diagram illustrating the function of E_2_ and P_4_ on the tight junction proteins showed that the targets were increased; therefore the tight junction proteins did not change.

**Table 1 animals-12-01404-t001:** The primer information for the qRT-PCR.

Gene	Accession Number	Primers		Product Size (bp)
		Forward	Reverse	
occludin	XM_018065677.1	CATTTTTGCCTGTGTTGCCT	AATCCCTTTGCCGCTCTTGG	186
claudin-1	XM_005675123.3	CCCAGTCAATGCCAGGTATG	TCTTTCCCACTGGAAGGTGC	168
claudin-3	XM_018041071.1	TGGCTGTGCACTATCGTGTG	CGAGTCGTACACCTTGCACT	156
ZO-1	XM_018066118.1	AGCAGACGCAGAAAACCATCA	TCTCCACGCCACTGTCAAACT	225
β-actin	NM_001009784.3	GGATGATGATATTGCTGCGCTC	TCTCCATGTCGTCCCAGTTGGT	248

## Data Availability

Not applicable.

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
