# Peer review of "Regulation of Tight Junctions by Sex Hormones in Goat Mammary Epithelial Cells"

_animals, 2022, doi:10.3390/ani12111404_

Round 1

Reviewer 1 Report

In revision to the manuscript entitled "Regulation of tight junctions by sex hormones in goat mammary epithelial cells".

Significance of the work: The authors concluded that sex hormones play a role in regulating the TJ in mammary glands in goats which can led to understanding a more mechanistic way of which sex hormones regulate  mammary gland development and milk lactation via cell-to-cell connections.

I believe this study is with a significance to the field, however i have some suggestion that could improve the quality of the manuscript.

  • Levels of sex hormones during pregnancy and lactation in relations to Tj components as protein and mRNA expression is a little confused for the reader, a graph is really required to simplify these relations and conclusions that the authors mentioned. In addition if there is a cartoon or diagram illustrating TJ would be also important.
  • Could the authors explains why they choose these specific days in pregnancy  and lactation to measure the hormones levels and TJ components (Reference or explanations).
  • It would be helpful to add one sentence after each result concluding it to make the follow up by the reader easy. (For example, at line 219, sex hormones levels are associated with different TJ protein components, this is just a suggestion).
  • could the authors use another word instead of youth like puberty or pre-puberty whatever appropriate.

Minor comments:

  1. Check abbreviations appear for the first time through the manuscript; some of them are
    • line 89 PBS is already abbreviated before
    • line 167 could you define PMSF
    • line 172 could you define TBST
  2. Please rephrase the sentence at lines 201 and 202.

Author Response

In revision to the manuscript entitled "Regulation of tight junctions by sex hormones in goat mammary epithelial cells".

Significance of the work: The authors concluded that sex hormones play a role in regulating the TJ in mammary glands in goats which can led to understanding a more mechanistic way of which sex hormones regulate mammary gland development and milk lactation via cell-to-cell connections.

I believe this study is with a significance to the field, however i have some suggestion that could improve the quality of the manuscript.

Response: Thank you for your precious comments and advice which are valuable and helpful for revising and improving our paper. We have revised the manuscript according to your suggestion, and our point-by-point responses are presented as follows.

Q1. Levels of sex hormones during pregnancy and lactation in relations to Tj components as protein and mRNA expression is a little confused for the reader, a graph is really required to simplify these relations and conclusions that the authors mentioned. In addition if there is a cartoon or diagram illustrating TJ would be also important.

Response: Thank you for your suggestion. Firstly, we have done experiments in vivo and in vitro. The levels of sex hormones during pregnancy and lactation in relations to TJ protein expression are in vivo, which we want to investigate whether there is a relationship between changes of serum sex hormones and TJ protein expression in mammary gland. To make it clear, we have added sentences in the results on page 5, lines228-230, the details are as follows:

These results demonstrated that from Pd137 to Ld31, the change of claudin-1 consistent with the change of E2 and P4, while occludin and ZO-1 varied with the change of E2 to P4 ratios.

Besides, it is really true as you suggested that a graph or cartoon describing the relationship between sex hormones and TJ components expression (mRNA and protein) in vitro would make it clear to the reader, we have drawn a schematic diagram in Figure 5, see the details in the resubmitted Figure 5 as follows:

Figure 5. The diagram illustrating the function of E2 and P4 on the tight junction proteins.    : represented the targets are increased;      : representing the tight junction proteins did not change.

Q2. Could the authors explains why they choose these specific days in pregnancy and lactation to measure the hormones levels and TJ components (Reference or explanations).

Response: Thank you for your question. The reason for choosing these specific days in pregnancy and lactation are as follows: firstly, the length for the pregnancy days of goat is around 150 days, and pregnancy day 91 (Pd91) and Pd137 are at the stage of mid-pregnancy and late-pregnancy, during which stages the levels of E2 and P4 are relatively steady and high. While on lactation day 4 (Ld4) and Ld3, levels of E2 and P4 decreased into lower levels in comparison with that on pregnancy day 91 and pregnancy day137. Secondly, during Pd91 and Pd137, TJ is forming and is open, while the milk on Ld4 to Ld31 is colostrum milk and regular milk, respectively, which may suggest the switch of closure to open of the tight junction. Thus, the reason for we choosing these days is to clearly evaluate the relationships changes of TJ proteins with of levels of hormones.

Q3. It would be helpful to add one sentence after each result concluding it to make the follow up by the reader easy. (For example, at line 219, sex hormones levels are associated with different TJ protein components, this is just a suggestion).

Response: Thank you for your valuable suggestion, we have added concluding sentences after each result, the details are as follows:

Lines 228-230: These results demonstrated that from Pd137 to Ld31, the change of claudin-1 was consistant with the change of E2 and P4, while occludin and ZO-1 varied with the change of E2 to P4 ratios.

Lines 274-275: These results indicated that E2 and P4 play different roles in regulating TJ mRNAs and that P4 dominates the upregulation of TJ mRNAs.

Lines 295-297: These results suggested that E2 and P4 accelerated the protein expression of TJ related targets, and that P4 played a dominant role in upregulating TJ proteins. 

Q4. could the authors use another word instead of youth like puberty or pre-puberty whatever appropriate.

Response: Thank you for pointing out our inappropriate description of animal reproductive cycle, we have changed youth into puberty in the manuscript and in all Figure 1, Figure 2, Figure 3, and Figure 4. See the details in the resubmitted manuscript and in figures.

Minor comments:

Q5. Check abbreviations appear for the first time through the manuscript; some of them are

line 89 PBS is already abbreviated before

line 167 could you define PMSF

line 172 could you define TBST

Response: we are very sorry for our negligence for writing abbreviations, according to your suggestion, we have checked the abbreviations and have defined the PMSF and TBST, see the details in the resubmitted manuscript on lines 101, 177, 181 and as follows:

protease inhibitor phenylmethanesulfonyl fluoride(PMSF)

Tris buffer solution added with tween-20 (TBST)

Q6. Please rephrase the sentence at lines 201 and 202.

Response: Thank you for your suggestion, we have made correction on the sentence on lines 201 and 202, see the details in the resubmitted manuscript on lines201-202 and as follows:

Original: “Finally after being washed with PBS again, the stained cells were visualized with EVOS fl auto imaging platform”;

Modification: “Finally, cells were washed with PBS and were imaged using an EVOS fl auto imaging system”.

Reviewer 2 Report

Where in mammary gland were tissues collected from?

some figures in Figure 4 needs some y-axis labels.

Would be nice to see some immuno done on the mammary gland tissue sections for the claudins and occuldins

Was qPCR done on the epithelial cells after culturing? If so, why not just do it on snap frozen tissue that you mentioned you collected?

Would also be nice to emphasize the limitations of doing all these experiments in vitro (in discussion)

Author Response

  1. Where in mammary gland were tissues collected from?

Response: Usually, we make a skin incision in the avascular area on the mid-abdominal line between two mammary gland and then collect 1cm3 mammary tissue in size which was 2-3 cm deep under the skin on one mammary gland. The mammary tissue in all groups were collected in the same side of the mammary gland using a surgical method under general anesthesia. To make it clear, we have added these sentences in the resubmitted articles on lines 89-92, the details are as follows: 

We make a skin incision in the avascular area on the mid-abdominal line between two mammary gland and then collected about 1cm3 mammary tissue in size which was 2-3 cm deep under the skin on one mammary. Mammary gland tissues in different groups were collected on the same side of the mammary gland using a surgical method under general anesthesia.

  1. some figures in Figure 4 needs some y-axis labels.

Response: Thank you for pointing out our mistakes, we have added the labels for Y- in the histogram for analyzing occludin and zo-1 in the resubmitted Figure 4. See the details in the resubmitted Figure 4 and as follows:

  1. Would be nice to see some immuno done on the mammary gland tissue sections for the claudins and occuldins

Response: Thank you for your suggestion, we have performed the immunofluorescence analysis for claudin-1, claudin-3, occludin, and ZO-1 in the goat mammary gland tissue collected on puberty, Pd91, Pd137, Ld4, and Ld31. Since there are some impurities in the tissue sections which may affect the picture effect, we have not arranged these parts in Figures, see the details as follows:

  1. Was qPCR done on the epithelial cells after culturing? If so, why not just do it on snap frozen tissue that you mentioned you collected?

Response: Thank you for your advice, in fact, we did not perform qPCR analysis directly after epithelial cells isolated, but instead, we performed qPCR and western blot analysis on the cultured epithelial cells administrated with different levels of hormones (E2 and P4). The snap frozen tissue was collected from the mammary gland of goats at the stage of puberty, Pd91, Pd137, Ld4, and Ld31. While in these stages, TJs may be regulated by several other hormones except E2 and P4. Thus, to evaluate single E2 or P4 effect on mRNA and protein expression of claudins, occludin, and ZO-1, we performed qPCR and western blot analysis on cultured epithelial cells in vitro after different levels of E2 and P4 administration. To make it clear, we have added one sentence in the results in the resubmitted manuscripts on Lines 299-301 and on Lines 320-322, see the details as follows:

Lines 265-266: To further explore the effects of E2 and P4 on TJ related targets, different levels of E2 and P4 were given to GMECs in vitro to assess the mRNA expression of TJ components.

Lines285-287: Then, to assess whether E2 and P4 also affect TJ proteins expression, the expression of claudin-1, claudin-3, occludin, and ZO-1 in GMECs treated with different levels of E2 and P4 were identified with western blotting analysis.

  1. Would also be nice to emphasize the limitations of doing all these experiments in vitro (in discussion)

 Response: Special thanks to your comment. In the current study, we performed the experiments both in vivo (mammary gland at the stage of puberty, Pd91, Pd137, Ld4, and Ld31) and in vitro (GMECs). It is true that there are some limitations of doing the experiments in vitro, thus we have added the limitations in discussion. See the details in discussion in the resubmitted manuscript on lines 371-377 and as follows:

In the current study, the experiments of administrating E2 and P4 on epithelial cells in vitro have many advantages such as relatively simple influencing factors and easy being controlling. However, since the internal environment of the body are regulated by complex network of nervous system, endocrine system and immune system, the in vitro cell experiments are difficult to simulate the environment in vivo and may not fully reflect the situation of the hormones on the mammary gland TJs expression. Thus,a further experiment in vivo is required for verifying the function of sex hormones on TJs.

Reviewer 3 Report

Dear authors,

Thank you for this piece of work, which i found very intriguing. I found undoubted merit in it, however, let me tell you that you should improve the presentation of the work. 

I believe that it needs an extensive language editing which, on the other hand, I do not believe should limit the evaluation of the scientific soundness of your paper. In any case, the manuscript cannot be published the way it is written, thus I warmly invite you to let it proofread for English grammar and style. Just for instance, a male goat is a buck. 

Another main point is that I believe that the introduction should be restructured. The paragraphs you presented, following each other, do not lead to a clear hypothesis nor state the objectives of your paper. You start very competently with the description of tight junctions and the selective role of the membrane modification,  as strongly influenced by mammary gland activity (blood/milk barrier). Which is crystal clear to me. But, as a reader, I found that the link is missing in what you describe from a molecular point of view and the hormonal crosstalk. The glue between the two is missing, which is, I feel, the Physiology of lactation. So, as in your premise you describe that a selective activity is carried out to keep apart blood and milk (which are fluids) and electrolytes make the difference in the body fluid distribution (mammary gland function increases blood flow to the district), I would suggest to introduce a short passage describing this physiological adaptation  at L. 47. For your convenience, I would suggest to report what found in goats by Cappai et al. (2019, Res Vet Sci, 123:84-90), as to electrolytes and body fluid distribution. Then you are expected to describe the hormonal asset (which you do). But here we come to the point: please 1)clearly state your hypothesis on the basis of gaps in the present literature and 2) clearly state the objectives of your investigation. In the last paragraph of introduction,  I would certainly remove the anticipation of material and methods. So, remove those and be very clear in assessing hypothesis and objectives and you can stop your introduction, in my opinion.

In M&M, please, provide details on the surgical removal of mammary tissue (techniques and dimension of tissue removed). I have not found the approval of the ethics committee, but you probably enclosed it as additional file.

Results are fine.

I like the discussion, but please avoid to be redundant with introduction in some passages.

Thank you.

.

Author Response

Thank you for this piece of work, which i found very intriguing. I found undoubted merit in it, however, let me tell you that you should improve the presentation of the work. 

Response: Special thanks to you for your good comments. Your suggestions and comments really helped us a lot.

  1. I believe that it needs an extensive language editing which, on the other hand, I do not believe should limit the evaluation of the scientific soundness of your paper. In any case, the manuscript cannot be published the way it is written, thus I warmly invite you to let it proofread for English grammar and style. Just for instance, a male goat is a buck. 

Response: Thank you for your suggestion. The manuscript has been edited and proofread by English professionals, the revised parts were highlighted with red colors, please see the details in the resubmitted manuscript.

  1. Another main point is that I believe that the introduction should be restructured. The paragraphs you presented, following each other, do not lead to a clear hypothesis nor state the objectives of your paper. You start very competently with the description of tight junctions and the selective role of the membrane modification, as strongly influenced by mammary gland activity (blood/milk barrier). Which is crystal clear to me. But, as a reader, I found that the link is missing in what you describe from a molecular point of view and the hormonal crosstalk. The glue between the two is missing, which is, I feel, the Physiology of lactation. So, as in your premise you describe that a selective activity is carried out to keep apart blood and milk (which are fluids) and electrolytes make the difference in the body fluid distribution (mammary gland function increases blood flow to the district), I would suggest to introduce a short passage describing this physiological adaptation at L. 47. For your convenience, I would suggest to report what found in goats by Cappai et al. (2019, Res Vet Sci, 123:84-90), as to electrolytes and body fluid distribution. Then you are expected to describe the hormonal asset (which you do). But here we come to the point: please 1) clearly state your hypothesis on the basis of gaps in the present literature and 2) clearly state the objectives of your investigation. In the last paragraph of introduction, I would certainly remove the anticipation of material and methods. So, remove those and be very clear in assessing hypothesis and objectives and you can stop your introduction, in my opinion.

Response: According to your suggestions, we have rewritten the introduction by introducing the physiology of lactation, the hypothesis and objectives of the study. We have also removed the anticipation of material and methods. See the details in the revised manuscript on lines 35-80 and as follows:

Mammary development begins in early fetal life, occurs slightly during puberty, while complete mammogenesis take place during pregnancy and become fully functional after parturition. All these steps of mammary gland development are tightly coordinated both by systemic hormones and local factors. Secretory differentiation of the alveolar mammary epithelial cells(MECs)starts around mid-pregnancy and become competent to produce and secrete some milk components at late pregnancy, while only until parturition, the milk secretion can be triggered [1]. The tight junction (TJ) is an important intercellular junctional complex which plays a vital role in controlling the epithelial barrier [2, 3]. In the mammary gland, TJ act as both a barrier and a fence: it could prevent paracellular transport of ions and small molecules across the cell bed (barrier function) as well as separate the cell membrane into distinct domains of protein and lipids (fence function).The mammary epithelium is leaky before lactation, while during the first days of lactation the bidirectional paracellular exchanges of molecules between the interstitial space and alveolar lumen is inhibited due to the closure of TJ triggered by the hormonal changes. From pregnancy to lactation, molecules can enter milk from depending on paracellular pathways into that of transcellular pathways to adapt to the high secretory state after parturition, these are due to the open or closure of tight junction (TJ) [4]. In the transition goats, the circulating electrolytes such as Na, Chloride, Mg, and K were assumed to change following the different distribution of fluids in the late gestation, birthing and early lactation [5], while loss of TJ integrity in goat mammary gland during lactation were closely related with milk reduction and the disturb of epithelia barrier [9-11]. Cows producing unstable milk also presenting elevated tight junction’s permeability [12][13].

Ovarian hormones of 17β-estrogen (E2) and progesterone (P4) are instrumental for mammary epithelial cell proliferation, ductal morphogenesis, mammary epithelium side branching and alveolar formation during puberty and pregnancy [14]. There was solid evidence that glucocorticoids, prolactin and steroids are required for formation and maintenance of mammary TJ. Studies whether by ovariectomy or by exogenous administration reported that P4 played an important role in preventing mammary TJ formation [4]. In amniotic epithelium, P4/progesterone receptor (PR) pathway maintains TJ by increasing claudin-3 and claudin-4 expression in a dose-dependent manner during midpregnancy [15]. Unlike P4, a vitro study on human primary gut tissues demonstrated that estrogen treatment decreased zonula occluden 1 (ZO-1) mRNA and protein expression which suggested that estrogen may play a role in gut tight junction expression and permeability [16]. In the female vagina, estrogen replacement could restore the decreased expression of ZO-1, occludin, and claudin-1 to the control level after ovariectomy [17]. These studies suggested the important role of estrogen and P4 on modulating the expression of TJ in different tissues. However, though TJ plays an important role in maintaining milk components in ruminant animals, there are rarely studies reported the expression pattern of TJ proteins and how ovarian hormones regulate TJ proteins which may affect lactation.

Since TJ structure is composed of transmembrane proteins of claudins and occludin as well as scaffolding proteins of zonula occludens (ZOs), TJ formation or structure may vary with change of these TJ protein expression. Thus, in the current study, we aimed to investigate the precise function of estrogen and P4 on the mammary TJ components expression during transition which may then affect TJ open or closure and finally lactation. The current research would be useful to clarify the mechanism of how E2 and P4 affecting lactation through regulating tight junction.

  1. In M&M, please, provide details on the surgical removal of mammary tissue (techniques and dimension of tissue removed). I have not found the approval of the ethics committee, but you probably enclosed it as additional file.

Response: Thank you for your suggestion, we have added the details on the surgical removal of mammary tissue according to your advice. Besides, the approval of the ethics committee was the Animal Care and Use Committee of Huazhong Agricultural University which has been included in the Materials and Methods. To make it clear, we have added and modified the sentences, see the details on the revised manuscript on lines89-92 and on lines 96-98, and as follows:

Lines89-92: Usually, we make a skin incision in the avascular area on the mid-abdominal line between two mammary gland and then collected about 1cm3 mammary tissue in size which was 2-3 cm deep under the skin on one mammary.

Lines 96-98: All animal experiments were carried out in accordance with and were approved by the Animal Care and Use Committee of Huazhong Agricultural University (ID number: HZAURA-2015-008).

  1. Results are fine.
  2. I like the discussion, but please avoid to be redundant with introduction in some passages.

Response: Thank you for your suggestion, we have modified some passages which may be redundant with introduction, see the details on page, lines 310-312 and as follows: 

Original passages: Several reproductive hormones were reported in regulation of TJ, such as E2 was involved in regulation of human gut or rat vagina tight junctions by affecting ZO-1, occludin, or claudin-1 expression [15, 16]; P4 maintains amniotic TJs by affecting localization and expression of claudin-3 and claudin-4 during midpregnancy in mice [14].

Modified passages: Reproductive hormones E2 and P4 were reported in regulation of TJ in the tissues of human gut ,rat vagina, or mice amniotic by affecting ZO-1, occludin, or claudins expression [15-17].
